# Health Workers’ Responses to COVID-19 Pandemic’s Impact on Service Delivery to Adolescents in HIV Treatment in Cape Town, South Africa: A Qualitative Study

**DOI:** 10.3390/healthcare12060609

**Published:** 2024-03-07

**Authors:** Yolanda Mayman, Talitha Crowley, Brian van Wyk

**Affiliations:** 1School of Public Health, University of the Western Cape, Bellville 7535, South Africa; bvanwyk@uwc.ac.za; 2School of Nursing, University of the Western Cape, Bellville 7535, South Africa; tcrowley@uwc.ac.za

**Keywords:** COVID-19, adolescents, HIV, antiretroviral therapy, healthcare systems, adherence, vaccine uptake, engagement, mental wellness

## Abstract

Adolescents living with HIV (ALHIVs) are considered a priority population in the fight against HIV, requiring dedicated services. The COVID-19 pandemic and subsequent disruptions deprived ALHIVs on antiretroviral therapy (ART) of the care and social support essential for treatment adherence and positive treatment outcomes. This study describes health managers’ and healthcare workers’ responses to the impact of COVID-19 on service delivery to ALHIVs in HIV treatment in the Cape Town Metropole. A descriptive qualitative design was employed, where semi-structured individual interviews (n = 13) were conducted with senior and programme managers as well as healthcare workers between April and October 2023. Inductive thematic analysis was performed using Atlas.ti version 23. Two main themes emerged from these interviews: “HIV service delivery to adolescents during the COVID-19 pandemic” and “Lessons learnt—the way forward”. The de-escalation of health services at primary health facilities and the disruption of HIV services resulted in disengagement from care by ALHIVs, increasing mental health and treatment challenges. This warrants the restoration of psychosocial support services and the re-engagement of ALHIVs. The findings from this study can function as a guide for health systems and healthcare providers to navigate future pandemics to ensure that vulnerable populations such as ALHIVs continue to receive care and treatment.

## 1. Introduction

In 2020, the COVID-19 pandemic impacted healthcare systems and facilities globally, as containment measures exacerbated existing challenges within healthcare systems [1,2]. The first COVID-19 case in sub-Saharan Africa was reported on 27 February 2020, and as of 8 November 2023, 102,595 COVID-19-related deaths had been reported in South Africa [3]. The pandemic and subsequent lockdown measures impeded access to clinical care, thereby delaying access to HIV treatment [4,5]. In July 2020, 73 countries reported an increased risk of antiretroviral therapy (ART) stockouts because of COVID-19 and a predicted 10% increase in HIV-related deaths in high-burden settings as a result of drug shortages and a lack of capacity in healthcare systems [2]. 

In 2020, the Joint United Nations Programme on HIV/AIDS (UNAIDS) released new 95–95–95 targets, calling for 95% of people living with HIV (PLHIV) to know their HIV status, receive sustained ART, and have viral suppression by 2025 [6]. The ART program in South Africa is the largest in the world, with more than 3.4 million individuals with HIV accessing and receiving antiretroviral (ARV) drugs [7]. Currently, 8.45 million people in South Africa live with HIV, which is an estimated 13.9% of the entire population [8].

It is reported that when compared to adults living with HIV (20 years and older) and children (under 10 years), ALHIVs experience significantly lower rates of retention, adherence, and viral suppression, as well as higher rates of interruptions to treatment [9,10,11,12,13]. In 2017, it was reported that in South Africa, 62.3% of ALHIVs knew their HIV status, 65.4% were on ART, and of these 78.1% had attained viral suppression [14]. This is notably lower than the targets set out by UNAIDS. The COVID-19 pandemic hindered the ability of adolescents and youths living with HIV to arrange scheduled appointments and manage regular treatment regimens [15]. An estimated 1.7 million children under 15 years of age were living with HIV in sub-Saharan Africa in 2020 [14]. Adolescents (10–19 years) are a particularly vulnerable population as they are in the critical stages of biological and social development, which complicates care delivery to this group [16,17,18]. Adolescents living with HIV (ALHIVs) often experience internal and external stressors related to mental health conditions. Stressors such as social stigma, feelings of isolation and shame, disclosure of their HIV status, and other behavioral difficulties negatively affect their adherence and treatment outcomes [9,19,20].

The COVID-19 pandemic adversely affected the delivery of HIV care and treatment to people living with HIV. COVID-19 restrictions due to the risk of transmission of the coronavirus hindered access to HIV care [21]. Evidence shows that these disruptions to treatment and services for PLHIV often lead to increases in mortality risk [21]. Additional barriers to HIV care and treatment services included a shortage in the supply of ART medication, a reduction in the provision of HIV services, and a shortage of resources [21,22]. Research also shows that COVID-19 has been associated with an increase in psychological distress [23,24]. Individuals with risk factors for HIV and who are vulnerable to mental health conditions often face other significant challenges to accessing and adhering to HIV prevention and treatment [25]. ALHIVs, a notably vulnerable population, were at increased risk of anxiety, depression and prolonged isolation during the COVID-19 pandemic as existing challenges faced by ALHIVs, such as concomitant stigmatization and normal adolescent development difficulties, were further exacerbated [26]. ALHIVs experienced a lack of access to clinic-based services such as counselling and peer-support groups during the pandemic [27]. This situation exposed adolescents to structural challenges, such as the absence of secure health facilities and parental requirements. This further imposed additional barriers to accessing essential healthcare services [16]. For ALHIVs, additional services such as psychosocial support are often needed to improve ART treatment outcomes.

Healthcare workers experienced increased pressure, workload and the reshaping of health service delivery, all attributed to the COVID-19 pandemic [28]. Due to this, healthcare workers can provide valuable insight into the impact of COVID-19 on the delivery of HIV services and the direct impact of the pandemic on health systems. The legacy of the COVID-19 pandemic therefore highlights the importance of ascertaining the experience of healthcare workers who provide HIV and ART services to ALHIVs. This paper describes the experiences of senior and programme managers and healthcare workers on the impact of the COVID-19 pandemic on the service delivery of treatment to ALHIVs in the Cape Town Metropole, South Africa.

## 2. Materials and Methods

### 2.1. Study Design

A descriptive qualitative study design was used. A qualitative study design is well suited when there is a need for a firsthand descriptions of the facts of a phenomenon and in-depth information on the experiences of informants [29,30].

### 2.2. Study Setting

This study was centered within the wider Khayelitsha eastern sub-district in the Cape Town Metropole in the Western Cape province of South Africa. According to the 2011 census (9–10 October 2011), Khayelitsha has a population of 391,749. In 2018, it was estimated that the population had grown to 500,000. The unemployment rate stands at 54.1%. In 2011, the official unemployment rate for people aged 15 years and above was 38%. The prevalence of HIV/AIDS is very high. In 2014, 31% of young women and 8% of men were HIV positive.

### 2.3. Sample Size and Sampling

In this study, 13 purposively sampled senior and programme managers as well as health workers who provide healthcare services to adolescents on HIV treatment were interviewedData saturation was reached after the 12th interview and a 13th interview was conducted to confirm data saturation. To be eligible for inclusion in the study, participants were required to be program managers or health workers in ART clinics within the Khayelitsha eastern sub-district, have experience working with ALHIVs on ART during the COVID-19 pandemic at healthcare facilities within the Cape Town Metropole, and be engaged in the delivery of ART services to ALHIVs during the mentioned period.

### 2.4. Data Collection

Individual in-person semi-structured qualitative interviews were conducted by experienced researchers between 5 April and 24 October 2023 at various locations within the Cape Town Metropole. An interview guide was used for the interviews, and questions such as “What is your role when it comes to delivering HIV care to adolescents with HIV?” were asked. The durations of the interviews were 45 min to 1 h, and before each interview the purpose of the interviews was discussed, and consent was sought from the participants. Interviews were conducted in English and were audio recorded. They were then transcribed verbatim and prepared for analysis.

### 2.5. Data Analysis

All interviews were uploaded to Atlas.ti and subjected to inductive thematic analysis. Thematic analysis facilitates the identification, analysis, and reporting of patterns or themes within a given dataset [31,32,33]. Inductive thematic analysis is data-driven and allows for interpretation of the various aspects of the research topic and the researcher’s interaction with the data [34]. Analysis began with the researcher engaging with the collected data through reading and ensuring familiarity with the data contained therein. Initial codes were then produced based on areas of commonality in the data, the identified codes were organized and categorized into potential themes, and the themes were refined. A detailed analysis of each of the themes was conducted, and a report on the analyzed themes was written.

### 2.6. Ensuring the Trustworthiness of This Study

Various criteria were developed to ensure trustworthiness and rigor [35,36]. These were credibility, dependability, confirmability, and transferability [35,37,38]. Credibility was achieved through the researchers asking clarifying questions, summarizing during interviews, and reflecting on their own experiences and engagement during this process through reflective notes and peer debriefing. Transferability was achieved through the research team providing an account of where and how the research was conducted, to ensure that the research can be applied in other conditions or with other groups. Verbatim quotes were also used to support interpretations of the findings.

In this study, dependability was achieved through the research team confirming and verifying the decisions made during each stage of the research process. Lastly, confirmability was achieved through reflexive journaling as well as the researchers explaining how conclusions and findings were established.

### 2.7. Ethics Considerations

Ethical clearance for this study was obtained from the University of the Western Cape Biomedical Research Ethics Committee (BM23/3/7). The researchers sought verbal and written consent from all included study participants. Numbers and codes were used to protect the identity of participants and maintain anonymity. Participants consented to the digital recording of interviews and the publishing of their responses if they were kept anonymous.

## 3. Results

### 3.1. Demographic Information

A total of 13 health managers and healthcare workers participated in this study (see Table 1). Participants consisted of 10 females and 3 males who occupied senior- (n = 4), mid- (n = 3), or implementation- (n = 6) level positions at primary healthcare facilities in the health district in the Cape Town Metropole.

Two main themes emerged from these interviews: “HIV service delivery to adolescents during the COVID-19 pandemic” and “Lessons learnt—the way forward” (see Table 2). These are further discussed below.

### 3.2. Theme 1: HIV Service Delivery to Adolescents during the COVID-19 Pandemic

#### 3.2.1. Challenges during the COVID-19 Pandemic

There were various challenges before and during the COVID-19 pandemic that impacted ALHIVs both directly and indirectly. Challenges such as a lack of space for adolescent-focused and youth-friendly services and inadequate financial resources to fund services for adolescents within facilities and staff existed pre-COVID-19, highlighting the lack of prioritization of care for ALHIVs. Additionally, there was a lack of dedicated mental health services and spaces for adolescents within clinics and secondary hospitals. The pandemic and the reallocation of space and healthcare workers exacerbated these existing challenges to managing the pandemic.


*“But finding a space in the day hospital that is relatively private is an ongoing nightmare. So it was always a bit of a tricky one, finding a place where we could have a group of people. Yeah, and then with COVID, obviously nothing could happen.”*

*(Participant 6, female, senior manager)*



*“Lack of space, lack of services, lack of prioritizing. Even here. We rarely admit adolescents for any mental health issues, because if they need to be they go straight to the adult ward. So if you are suicidal and we feel like they know this child cannot go home and they are quite at risk you get put in the adult ward and a lot can go wrong. Then to get them transferred like to any tertiary facility. It’s very difficult.”*

*(Participant 8, female, healthcare worker)*


As a result of lockdown measures and regulations, the treatment adherence of adolescents was affected due to the loss of support from their peers and other ALHIVs within facilities, as well as isolation from their peers at school.


*“So they lost the support that they normally get from their peers as well as from the facilities. So that might have affected their adherence to treatment as well as remaining virally suppressed.”*

*(Participant 12, female, healthcare worker)*


Youth clubs for adolescents and youth living with HIV were not fully implemented within all health facilities before the COVID-19 pandemic due to existing challenges. During the pandemic, youth clubs ceased and ALHIVs lost the support they received within these clubs. Post-COVID, many of these clubs have not resumed due to staff and space shortages.


*“We had a youth group and that’s something else. That is where we would sit down and talk about hygiene, for instance, or disclosure. Yeah, we had that before COVID, and we don’t have space now.”*

*(Participant 7, female, healthcare worker)*


Mental health issues and challenges increased during the COVID-19 pandemic. For ALHIVs, not being able to interact and socialize with peers, and experiencing the loss of their parents due to COVID-19 were circumstances in which adolescents required mental health support. However, these support services were not prioritized.


*“Yes, just knowing from my adolescents at home it was frustrating for them just to be stuck indoors. So yeah, you know that it does have mental health issues. Yeah, it will affect them negatively if they can’t go and socialize with their peers, and they also cannot just go out and have fun and play outside and with their friends.”*

*(Participant 12, female, healthcare worker)*



*“Obviously, teenagers saw their parents dying, that kind of stuff. So obviously mental health wasn’t prioritized like the normal things.”*

*(Participant 13, male, senior manager)*


#### 3.2.2. HIV Support Services

In an effort to provide HIV services to children and adolescents during lockdown periods, some healthcare workers implemented alternative support measures such as telephonic and virtual consultations within certain facilities, to maintain contact with ALHIVs. These healthcare workers found it necessary to monitor ALHIVs who were at an increased risk during the lockdown periods.


*“I’m saying we had to for some children switch to either telephonic consultations or to virtual, you know, zoom consultations, which is what we used a lot.”*

*(Participant 11, female, mid-level manager)*


Parent and caregiver workshops were also initiated to provide guardians with information on how to establish a supportive structure for adolescents within the home during lockdown periods.


*“And one of the things that we were also running were parent/caregiver groups and workshops that we had to switch to virtual. We shared a lot of material with them, shared a lot of advice with the parents on how to now set up a structure, you know, for that child, because now you are the caregiver.” *

*(Participant 11, female, mid-level manager)*


Online skills clubs were also started during the COVID-19 pandemic to help adolescents and youths with HIV to readjust to going back to school and to manage challenges within the home and school environments.


*“So then I initiated like having online groups. So the teens would have access and I would send you a link once a week and they could join the group and then we would run an hour group, usually like from 3 to 4, like after school hours. And that was mainly like skills groups like teaching them how to cope with like difficult things in their lives.”*

*(Participant 8, female, healthcare worker)*


The cessation of health services within facilities during COVID-19 meant that HIV support services were less accessible for ALHIVs. Alternative measures such as virtual and telephonic consultations in certain facilities meant that ALHIVs who were deemed at-risk had the opportunity to be engaged in care and with individual healthcare workers. The challenges relating to the delivery of HIV services and care to ALHIVs that existed before the pandemic were further exacerbated during the COVID-19 pandemic.

### 3.3. Theme 2: Lessons Learnt—The Way Forward

#### 3.3.1. Comprehensive Health Services for ALHIVs

Increasing the mental health focus through awareness and consistent screening was highlighted as one of the ways to improve the mental health services available to ALHIVs within healthcare facilities.


*“Yeah, I think from our department I think one of the awareness that is raised that we’ve also tried to put out there, especially to caregivers and to society is an understanding that children and adolescents do get affected as much as adults by mental illness.”*

*(Participant 11, female, mid-level manager)*



*“I’m not sure that I would say COVID taught this to me, but it reinforced what I have always thought that peer and mental support is just really important for these teenagers. And COVID was very isolating.”*

*(Participant 6, female, implementation-level manager)*


The implementation of youth-friendly services in facilities was emphasised. By providing a space for adolescents and youth within healthcare facilities, health-seeking behaviour and healthcare engagement can also be improved. Appropriate and youth-friendly staff members must also be trained and appointed to provide youth-related services within health facilities.


*“It’s just their space. It’s very important for them. And also, the services must be delivered by someone who understands the young people. Someone to whom they can relate. A young nurse. You know, we speak about youth-friendly services, but some facilities, you’ll find that the nurse who is at the youth clinic is actually not youth-friendly. So just besides the space as well the staff in the environment must also compliment the services so that young people feel comfortable to access those services.”*

*(Participant 12, female, healthcare worker)*


#### 3.3.2. Healthcare System Future Preparedness

The COVID-19 pandemic highlighted the need for responsiveness and a more proactive response from the Government. This involves being prepared for future pandemics.


*“Government services are generally crisis orientated. We generally don’t plan for how we can provide a better service for you in the future. We’re not forward looking. We provide the service as clients appear.”*

*(Participant 1, male, senior manager)*


Telehealth and medication-delivery systems are alternative consultation and medication-delivery options that were implemented during the pandemic to ensure the continuation of patient treatment. It is recommended that these systems continue, to further improve patient consultation and the delivery of medication to those who are unable to access facilities.


*“Yeah, I think they had great ideas, and I don’t know why they didn’t continue with it. Yes, cause they started with this mailbox. So there was mailboxes outside. Delft had a system where they delivered their medications, but everything has stopped now and that’s actually very effective.”*

*(Participant 10, female, healthcare worker)*


Another recommendation on how to ensure that the healthcare system is prepared is to have an excess supply of medication within facilities. This will ensure that patients are still able to receive their medication and maintain their treatment adherence during a future pandemic.


*“Always expect the unexpected. Have prepacked medication with long term expiry dates ready. Have enough stock and supply of everything that you might need because we don’t know what else can come up tomorrow.”*

*(Participant 12, female, healthcare worker)*


Community-based HIV training should be focused on and implemented to enhance the dissemination of knowledge beyond healthcare facilities and provide better assistance during the disclosure of HIV status. This approach could address issues of stigma within communities and assist healthcare workers who provide support to ALHIVs within facilities.


*“They [parents] need a better education because we do our disclosures and stuff ourselves and we start by 9. Most parents don’t want from. They will say maybe 11 to 12, maybe older, but if you start talking to them, their knowledge of HIV is actually zero.”*

*(Participant 10, female, healthcare worker)*


## 4. Discussion

The findings of our study demonstrate the impact of the COVID-19 pandemic on the HIV treatment and mental well-being of ALHIVs. Previous studies have shown that during the COVID-19 pandemic, HIV-related health services were notably reduced due to the reallocation of staff and resources to managing COVID-19 outbreaks and services [39,40]. The COVID-19 pandemic exacerbated existing challenges within healthcare facilities, thereby affecting HIV care and service delivery to ALHIVs. Shortages of staff, space, and financial resources affected the prioritization of ALHIVs. Researchers state that pre-pandemic adherence patterns among ALHIVs were already worse than that of other groups living with HIV, due to a lack of adequate psychological support for children and adolescents [11,29,41]. Additionally, viral non-suppression amongst children and adolescents on ART were also caused by individual factors such as incomplete treatment adherence, missed clinic appointments as a result of financial challenges, and interruption of access to treatment and food interactions [41]. This placed them at a higher risk of poor adherence vulnerability during the COVID-19 pandemic [42].

Mental health challenges and difficulties faced by ALHIVs increased during the COVID-19 pandemic as a result of the inability to socialize with peers and the loss of loved ones. Research conducted in South Africa shows that mental and emotional challenges amongst ALHIVs during the pandemic were linked to depression, anxiety, stress, social isolation because of the lockdown, movement restrictions, and the loss of loved ones [27,43,44]. The measures implemented by authorities during the COVID-19 pandemic gave rise to additional psychological and mental stressors, leading to an increased risk for psychological illness amongst vulnerable groups such as ALHIVs [42,45]. This is particularly concerning as adolescence is an important period of critical development that determines the mental health and overall well-being of an adolescent [46,47]. Findings from this study found that there is an increased need for the prioritization of mental health support services and screening for ALHIVs.

Lessons learnt from the COVID-19 pandemic show that the cessation of certain health services within healthcare facilities during the pandemic introduced alternative support measures aimed at providing continued support services to ALHIVs. Telephonic and virtual consultations, together with caregiver workshops provided by certain healthcare workers in this study, proved effective as ALHIVs deemed at-risk received continued individual care. Telemedicine, which is a form of telehealth aimed at providing clinical services, was applied within facilities to provide access to HIV care and treatment during the COVID-19 pandemic [41]. Evidence shows that employing telehealth with HIV care has several advantages, which include improving access to care and reducing stigma-related delays in care [48]. Further research reveals that the use of telehealth and virtual interventions in programs were effective in facilitating HIV medication delivery and testing and improving access to antiretroviral therapy when physical services were not deemed feasible [48,49]. Results from a study conducted in the United States of America found that both patients living with HIV and healthcare providers reported positive attitudes towards and experiences with telemedicine during the COVID-19 pandemic [41]. Increased efforts to maintain regular contact with ALHIVs by means of texting, email, phone calls, and virtual activities were critical to ensuring treatment retention and engagement during the COVID-19 pandemic [49]. It is suggested that pandemic-related service disruptions, which include drug supply delays and shortages, impeded the continuation of HIV treatment and services [42]. A study in South Africa found that primary healthcare systems such as home delivery and telehealth are critical to improving access to care and the efficiency of services [50]. Therefore, measures such as telehealth and medication delivery systems should be continued post pandemic, as well as an excess supply of medication within facilities.

Our study findings confirm that HIV-related stigma remains an issue that affects care and engagement in the treatment of people living with HIV. A study conducted in South Africa found that internalized stigma ranged from 22% to 41%, while the prevalence of a stigmatizing experience ranged from 43.5% to 88% [51]. Evidence from this present study suggests that anticipated and experienced stigma within communities are barriers to healthcare access and treatment adherence for people living with HIV. Research shows that high and persistent levels of anticipated stigma may result in non-adherence to ART, disengagement in care, and non-disclosure amongst ALHIVs [51,52,53,54]. Additionally, a study conducted in Rwanda found that anticipated stigma was common amongst PLHIV, as it was sometimes experienced within communities and health facilities, often leading to early disengagement from care [55].

Participants in this study recommended that an excess supply of medication is needed within healthcare facilities as part of future preparedness. A study conducted in South Africa found 13% of people living with HIV reported not having access to their medication during the COVID-19 pandemic [56]. This may lead to an increase in virological failure and vulnerability to infections [56]. It is thus important that the healthcare system takes the lessons learnt during the COVID-19 pandemic into consideration when formulating health intervention strategies and alternative methods of chronic medication dispensation, improves communication across healthcare platforms and within communities, and improves the use of telehealth to avoid the threats of possible future infectious disease outbreaks. This kind of preparedness in the healthcare system may ensure that patients can receive their medication and maintain adherence during a pandemic.

### Study Limitations

Due to the qualitative nature of this study, there are limitations. Firstly, participants were recruited based on their role as program managers or health workers in healthcare facilities within the Cape Town Metropole, as well as their engagement in the delivery of ART services to ALHIVs during the COVID-19 pandemic. Secondly, due to the differences in the delivery of care to ALHIVs across health facilities, the experiences and perspectives of participants in this study may not represent those of other participants. However, the information from this study may be used for future research aimed at further exploring the impact of the COVID-19 pandemic on ALHIVs in South Africa.

## 5. Conclusions

While adolescents and youth are at a lower risk of COVID-19 infection and mortality, containment and restrictive measures to combat the COVID-19 pandemic introduced challenges and opportunities in the delivery of HIV care and support services for ALHIVs. The disruption in HIV support services as a result of the pandemic calls for the continuation of HIV support services tailored to ALHIVs as well as the restoration of alternative support measures to recover from the losses inflicted by the COVID-19 pandemic. The lessons learnt can function as a guide for health systems to navigate future pandemics and to ensure that vulnerable populations such as ALHIVs continue to receive the necessary care and treatment.

## Figures and Tables

**Table 1 healthcare-12-00609-t001:** Summary description of participants (N = 13).

Participant Number	Designation	Gender
1	District manager	Male
2	HIV programme manager	Male
3	HIV programme deputy director	Female
4	Medical officer	Female
5	Medical officer	Female
6	Medical officer	Female
7	Medical officer	Female
8	Clinical psychologist	Female
9	Social worker	Female
10	Medical doctor	Female
11	Clinical program coordinator	Female
12	Counselling coordinator	Female
13	Programme manager	Male

**Table 2 healthcare-12-00609-t002:** Description of themes, sub-themes, and codes.

Theme	Sub-Theme	Codes
HIV service delivery to adolescents during the COVID pandemic	Challenges during the COVID-19 pandemic	Lack of space
		Consequences of limited health services
		HIV medication adherence
		Lack of youth club implementation within facilities
		Cessation of youth clubs
		Increase in mental health challenges
	HIV support services	Telephonic and virtual consultations
		Parent/caregiver workshops
Lessons learnt—the way forward	Comprehensive health services for ALHIVs	Increased mental health focus
		Implementation of youth-friendly services in facilities
	Health system future preparedness	Medication delivery service
		Excess medication supply
		Community HIV education

## Data Availability

The data for this study are available upon request from the principal and corresponding author, Y.M.

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
