# Peer review of "Health Workers’ Responses to COVID-19 Pandemic’s Impact on Service Delivery to Adolescents in HIV Treatment in Cape Town, South Africa: A Qualitative Study"

_healthcare, 2024, doi:10.3390/healthcare12060609_

Round 1

Reviewer 1 Report

Comments and Suggestions for Authors

Comments are in attached file.

Comments on the Quality of English Language

Moderate editing is required.

Reviewer 2 Report

Comments and Suggestions for Authors

Thank you for an opportunity to review this manuscript. The study is relevant since it looks at the effect of COVID-19 on HIV care among ALHIV.

Introduction: The introduction was well written since it provides the reader with information on what is already known about the topic. This section is also well-referenced.

Materials and Methods: The authors clearly stated the study design and justified why it was chosen. The sample size and sampling methods were provided as well as the justification for the sample size. The authors clearly described how data were collected, data analysis, and the strategies that were taken to ensure the trustworthiness of the study.

Results: The results were presented logically. Quotations from the participants were given to support the sub-themes and the themes.

Discussion: The authors compared their findings with those of previous studies and gave explanations to their findings.

Conclusion: The conclusion is based on the study findings.

Reference List: The reference list adheres to journal requirements.

There are few errors that may need to be addressed to strengthen this manuscript.

1. In line 22, the authors state, ‘Service delivery and support to ALHIV on ART was severely disrupted during…’ Replace ‘was’ with ‘were’.

2. In line 32, the authors state, ‘….of 8 November 2023 102,595 COVID-19 related deaths have been reported in South Africa.’ Replace ‘have’ with ‘had’.

3. In line 94, the authors state, ‘This study is centered within the wider Khayelitsha Eastern sub-district in the Cape…’ Replace ‘is’ with ‘was’

4. Line 157-159 do not provide demographic information and should be removed from this sub-heading.

5. In line 277, the authors state, ‘….partly to youth organizations not wanting to be involved in community mobilization.’ Add ‘due’ after ‘partly’

6. In line 390, the authors state, ‘Shortages in staff, space and …’ Replace ‘in’ with ‘of’

7. In Line 444-446, the authors state, ‘Evidence from this study suggests that anticipated and experienced stigma within communities are barriers to the access of health care and treatment adherence of people living with HIV [55,56,57].’ It is not clear whether this evidence is from their study or the other studies referenced. I suggest they state that these are findings from their study, then they state that similar findings were revealed in other studies, and they reference these studies.

Comments on the Quality of English Language

Minor editing required as stated above

Reviewer 3 Report

Comments and Suggestions for Authors

See the attached file for the detailed review report 

Comments on the Quality of English Language

Round 2

Reviewer 1 Report

Comments and Suggestions for Authors

Thanks for submitting the revision; my comments have been addressed, and the manuscript has been improved. I have no further comments.

Author Response

Dear reviewer 

Thank you so much for your feedback. It is highly appreciated!